# Cytoplasmic Mixing, Not Nuclear Coexistence, Can Explain Somatic Incompatibility in Basidiomycetes

**DOI:** 10.3390/microorganisms9061248

**Published:** 2021-06-08

**Authors:** Ben Auxier, Karin Scholtmeijer, Arend F. van Peer, Johan J. P. Baars, Alfons J. M. Debets, Duur K. Aanen

**Affiliations:** 1Laboratory of Genetics, Wageningen University and Research, 6708 PB Wageningen, The Netherlands; fons.debets@wur.nl; 2Mushroom Group, Plant Breeding Department, Wageningen University and Research, 6708 PB Wageningen, The Netherlands; Karin.Scholtmeijer@wur.nl (K.S.); arend.vanpeer@wur.nl (A.F.v.P.); Johan.Baars@wur.nl (J.J.P.B.); 3CNC Grondstoffen, P.O. Box 13, 6590 AA Gennep, The Netherlands

**Keywords:** basidiomycete, allorecognition, nonself recognition, somatic incompatibility, vegetative incompatibility

## Abstract

Nonself recognition leading to somatic incompatibility (SI) is commonly used by mycologists to distinguish fungal individuals. Despite this, the process remains poorly understood in basidiomycetes as all current models of SI are based on genetic and molecular research in ascomycete fungi. Ascomycete fungi are mainly found in a monokaryotic stage, with a single type of haploid nuclei, and only briefly during mating do two genomes coexist in heterokaryotic cells. The sister phylum, Basidiomycota, differs in several relevant aspects. Basidiomycete fungi have an extended heterokaryotic stage, and SI is generally observed between heterokaryons instead of between homokaryons. Additionally, considerable nuclear migration occurs during a basidiomycete mating reaction, introducing a nucleus into a resident homokaryon with cytoplasmic mixing limited to the fused or neighboring cells. To accommodate these differences, we describe a basidiomycete model for nonself recognition using post-translational modification, based on a reader-writer system as found in other organisms. This post-translational modification combined with nuclear migration allows for the coexistence of two genomes in one individual while maintaining nonself recognition during all life stages. Somewhat surprisingly, this model predicts localized cell death during mating, which is consistent with previous observations but differs from the general assumptions of basidiomycete mating. This model will help guide future research into the mechanisms behind basidiomycete nonself recognition.

## 1. Introduction

Nonself recognition is a prerequisite for the resilient mycelial network that makes up a fungal individual. This network is built by a set of growing hyphal tips that explore their environment in search of resources. Eventually, this growth results in the meeting of two hyphae, whether originating from the same individual or not. If these tips are from the same individual, then fusion will increase the efficiency of the network [1], while if the hyphae are from different individuals, fusion may permit the spread of parasitic elements [2,3,4,5]. In fungi, despite these apparent risks, conspecific hyphal interactions often result in fusion. Identity is then assessed post-fusion and if nonself is recognized the fusion cell is degraded, a process termed somatic incompatibility (SI). In the absence of nonself cellular markers, indicating a self-fusion, a stable fusion cell is formed as part of the interconnected hyphal network. Without such a nonself recognition mechanism, post-fusion continuation would be permitted between different individuals resulting in the risk of spreading elements like viruses and nuclear or mitochondrial parasitic variants. Nonself recognition followed by cell death reduces these risks but comes with the cost of the hyphal death, as well as reductions in spore production due to a reduced colony size [1]. To minimize the risks of building a hyphal network while allowing the benefits of fusion, a robust and precise system of nonself recognition is essential.

Systems of nonself recognition are not unique to fungi. Other multicellular organisms such as plants and animals also have such systems, often built on similar molecular pathways [6]. This maintenance of cellular identity is a wider phenomenon, also found in bacteria, and is likely shared among all cellular life [7,8]. Thus, even without a multicellular lifestyle, organisms still appear to benefit from preventing the spread of infectious agents, as seen in bacteriophage recognition systems [9]. In these systems, bacteria produce “self” markings on endogenously produced DNA. This allows a bacterium to discriminate against incoming DNA, such as from viruses since they lack these marks. Interestingly, in bacteria, these systems are also somewhat imprecise, and occasionally self-DNA is cleaved [10]. Such self-cleavage in bacteria shows that the costs of nonself recognition systems are common, but the benefits of these systems apparently outweigh the costs.

In basidiomycete fungal species, a SI phenotype is consistently observed between heterokaryons [11]. Thus, a lack of SI between isolates is generally used to identify clonal isolates. Culture-based SI observations are more practical than the use of genetic markers to identify clones and are thus widely used [12,13,14]. The macroscopic phenotypes of SI in basidiomycetes vary greatly, but SI is generally found between heterokaryons, although occasional reports of macroscopic responses between homokaryons have been mentioned [15]. Despite its repeated use as a tool for experimental mycology, genetics remains unclear [16,17,18,19,20], and the mechanisms remain completely unknown [11]. Here, we attempt to translate what is known about nonself recognition from ascomycete systems onto basidiomycete biology and explore the consequences of such mechanisms.

## 2. Genetics and Principles of Ascomycete SI

Typically, two randomly selected ascomycete individuals from a wild population will show SI following anastomosis. The known molecular determinants of nonself recognition in ascomycete systems are generally based on direct protein interactions [21]. Since ascomycete species are predominantly homokaryotic (i.e., with only one type of nucleus) and have haploid genomes, for any particular gene the products of only a single allele will be found in the cytoplasm. In these organisms, nonself recognition then results from interactions between the products of two alleles of the same gene (allelic; Figure 1 left, or between products of generally tightly linked non-homologous genes (non-allelic; Figure 1 right). Whether allelic or non-allelic, the general principle of ascomycete systems seems to be that for each locus, the cytoplasm of an individual will contain products from a single allele and upon fusion, mixing of two cytoplasms may result in the interaction of these products. The signal from this nonself-recognition is then transmitted to trigger cell death, often based on extremely conserved mechanisms [22]. While there are multiple loci where differences result in nonself recognition, they are not equivalent as the phenotypes can differ greatly both in macroscopic morphology [23] as well as efficacy to prevent the spread of parasitic elements [4]. Regardless of the specifics, the resulting cell death breaks the cytoplasmic connection between the two mycelia preventing, or at least limiting, the spread of parasitic elements. Within a population with multiple loci where differences confer incompatibility, most individuals will differ at a minimum of one locus. Thus, the fusion between two randomly selected individuals will mix these cytoplasms, and incompatibility between any two individuals can be expected [24].

However, cell fusion between genetically different ascomycete homokaryons should not always result in SI, as fusion is necessary during sexual reproduction. In ascomycete species following fertilization both haploid nuclei briefly co-exist in a heterokaryotic state until meiosis. During this stage, clearly cell death should be prevented despite the presence of two genomes. One possible solution is to reduce the concentration of incompatible proteins by dilution of the cytoplasm of one partner. Mating in ascomycete fungi is typically anisogamous since fertilization often involves male spermatia/conidia donating a nucleus with limited cytoplasm, preventing cytoplasmic mixing and subsequent lethal protein-protein interactions. A second solution is the active toleration of otherwise lethal protein combinations. In several *Neurospora* species, the mating locus itself acts as a nonself recognition locus, dependent on the TOL protein. During fruiting body initiation, TOL is downregulated in the sexual structures, allowing both mating types to be co-expressed within specific tissues [25,26].

Based on the above concepts, a population of an ascomycete species will be polymorphic at a number of loci, either allelic or non-allelic, with each locus being capable of triggering nonself recognition, resulting in SI. The number of polymorphic loci has been experimentally determined for only a few species, but it appears that there can be between 5 and 11 loci responsible [24]. However, it should be noted that many studies have recognized nonself responses based on macroscopic phenotypes, and recent experiments have identified loci involved in nonself recognition without a macroscopic phenotype [22,27]. The allelic variants of genes conferring nonself recognition specificity are generally found in even allele frequencies and with trans-species polymorphisms, general signs of balancing selection [28,29,30,31]. This form of selection occurs when the fitness value of an allele is high at low frequency and decreases as the allele becomes more common in a population as its utility for distinguishing self from nonself decreases [32,33]. The increased fitness of any rare allele leads to stable and even allele frequencies across populations [29]. Whether or not the selection derives from a role for nonself recognition genes in a fungal immunity system [6,34], or other sources of extrinsic selection such as mating loci [35,36], the resulting balancing selection leads to alleles being regularly shared between closely related species [29,30]. However, the polymorphic genes triggering nonself recognition are not regularly shared between higher groups of taxa. This means that a “core” set of incompatibility genes for all fungi does not exist, although there are domains that are repeatedly found to be involved [37]. This then implies that the mechanisms of nonself recognition are constantly evolving, and the ongoing recruitment of new proteins allows for greater specificity. This lack of a core set of ascomycete nonself recognition genes is consistent with previous research which failed to recover orthologs of ascomycete nonself recognition genes in basidiomycete genomes [38]. While basidiomycete nonself recognition may not involve orthologous genes, we may still be able to apply principles learned from studies in ascomycete species.

## 3. Differences between Ascomycete and Basidiomycete Species Regarding Nonself Recognition

Species of basidiomycetes, particularly the mushroom-forming groups, present a challenge for the existing paradigm of fungal nonself recognition. These fungi alternate between a monokaryotic state formed after meiosis and a persistent heterokaryotic state formed from the fusion of two homokaryons, in addition to new outgrowth being heterokaryotic. This outcrossing is facilitated by the extraordinarily high numbers of mating types [39], ensuring almost certain sexual compatibility [40]. It should be noted that while the sexual compatibility system is sometimes referred to as a “nonself recognition system” [41], the sexual system controls nuclear migration (amongst others) and is independent of the cell-death inducing nonself recognition discussed here. While not common, there are examples of homokaryons that have compatible mating loci yet appear to be either sexually incompatible or asymmetrically compatible [15,42]. An important but seemingly underacknowledged observation is that even in sexually compatible matings, cell death in the fusion cell is observed [43]. This cell death between homokaryons has not only been observed at the microscopic level but has been also observed macroscopically, as seen in *Rhizoctonia solani* [44,45].

To achieve near-universal sexual compatibility, any somatic incompatibility between the homokaryons must be overcome [46]. The compatibility of two distinct homokaryons means that nonself recognition must either be turned off when homokaryotic, allowing free fusion, or some other mechanism must allow the migrating nuclei to escape cell death. Following the fusion of two homokaryons, nuclei from each homokaryon reciprocally fertilize the other homokaryon, and rapid nuclear migration leads to both homokaryons becoming heterokaryotic themselves, in addition to new outgrowth being heterokaryotic [47]. Thus, this heterokaryon has two different genomes in a shared cytoplasm (Figure 2b). Ascomycete fungi also form a heterokaryotic stage but this is short-lived and/or occurs in specialized sexual tissues prior to ascospore formation. One solution found in *Neurospora* sp. to tolerate the brief heterokaryotic phase is through suppressing the action of the *tol* gene product, which otherwise triggers nonself recognition due to the mating-type alleles [26]. This suppression of mating-locus-associated SI is feasible because the ascomycete heterokaryon is short-lived, and reduced *tol* expression can be restricted to a specific time and place in the sexual cycle. This reduced *tol* expression, in theory, permits the spread of parasitic elements, however, the production of the specialized female structures (e.g., perithecial walls and paraphyses) prior to fertilization limits the potential for the spread of these parasitic elements. It has been shown in *Aspergillus nidulans* that the fertilizing nucleus can spread to the Hülle cells, accessory multinucleate cells, but only when strains are somatically compatible, indicating that SI limits this spread [48,49]. As the heterokaryotic stage is the persistent stage in most basidiomycete species, a system analogous to *tol* is not feasible, since prevention of recognition during the heterokaryotic stage is not compatible with the fact that incompatibility occurs between heterokaryons.

In addition to heterokaryon formation during basidiomycete matings, models of nonself recognition must be compatible with matings between heterokaryons and homokaryons, so-called Buller matings [50,51]. These involve the fusion of dikaryotic and monokaryotic hyphae, migration of nuclei from the heterokaryon into the receiving homokaryon, with eventual fertilization of the homokaryon by a single nuclear type from the original dikaryotic partner. These matings are perhaps common in nature [52] and do not seem to be affected by SI. Such a Buller mating produces a fusion cell containing three different nuclei, with nuclear migration from the heterokaryon into the receiving homokaryon (Figure 2c). Interestingly, it is known that SI can result from interactions between heterokaryons constructed to share a “common nucleus”, with one heterokaryon having nuclei A + B and the other A + C, nucleus A being common to both [11,16]. It seems that a fusion cell in a Buller mating, having three different nuclei, has a different outcome than the pairing of two heterokaryons that have a common nucleus, also having three different nuclei. Thus, the difference between a Buller mating and a “common nucleus” heterokaryon pairing cannot be due to genetics, as both involve three nuclear types in the fusion cell. Instead, perhaps the success of Buller matings relates to nuclear migration, which does not occur in pairing between heterokaryons with a common nucleus (Figure 2d). Despite the apparent differences between ascomycete and basidiomycete fungi, we argue that the presence of nuclear migration in basidiomycete species may explain a difference in outcomes but with similar underlying mechanisms.

## 4. Dikaryotic Ascomycete Fungi May Serve as a Guide to Understand Basidiomycete SI

Somatic incompatibility in ascomycete fungi generally occurs between homokaryons. However, in some pseudohomothallic ascomycete species, incompatibility also functions between heterokaryons. Isolates of *Neurospora tetrasperma* and *Podospora anserina* have nuclei of both mating types, and in these species, SI occurs between the stable heterokaryons, between homokaryons, as well as between heterokaryons and homokaryons. As long as incompatible protein products are expressed, somatic incompatibility results following hyphal fusion. Due to the pseudohomothallic lifecycle (i.e., spores are heterokaryotic for the mating type and the result of selfing), field isolates of these species are, as a rule, heterokaryotic and highly inbred, with little or no het-gene differences within a heterokaryon, and interactions in nature between homokaryons are likely rare.

As *N. crassa* is a primary ascomycete model for fungal nonself recognition, it seems logical to first examine the closely related *N. tetrasperma*, which remains heterokaryotic throughout its entire lifecycle. Members of this species produce heterokaryotic conidia, as well as producing four heterokaryotic ascospores per ascus instead of the standard eight homokaryotic ascospores. Thus, the two genomes reside in the same cell continuously, although homokaryotic ascospores are produced at a low rate. It is then interesting to consider how these organisms handle the complications of incompatibility from the co-habitation of two genomes. *N. tetrasperma* avoids the problem of mating-type associated SI by having inactive tol alleles [53]. Investigation of wild isolates has shown them to be homozygous at all other known het loci [54]. Furthermore, sexual crosses between homokaryons derived from strains that show incompatibility as heterokaryons (thus homokaryons that differ at het loci) produce heterokaryotic offspring that are self-incompatible, resulting in extremely poor growth [55,56]. This limits outbreeding in this species, as outcrossing produces offspring heterozygous for het loci which restricts growth severely [55]. Interestingly, the severe growth defect of self-incompatibility from heterozygosity at a het locus can be escaped, as mutations in one of the copies of a heterozygous het locus will restore wild-type growth and thus have a strong advantage [54]. Thus, in *N. tetrasperma*, SI between homokaryons is not fundamentally different from SI between heterokaryons.

The other model system for ascomycete SI, *P. anserina*, may also inform a model for basidiomycete nonself recognition. Several incompatibility systems from this species have been characterized, particularly the prion-forming *het-s* [57]. This system has two alleles: *het-S* which contains a HeLO domain that when exposed causes holes in the plasma membrane, and *het-s*, which contains a harmless HeLO domain [57] Figure 1. The HET-s* protein from the *het-s* allele can transition from a harmless non-prion form into the HET-s prion form which can trigger the HET-S protein to cause death [58]. Homozygous *het-s/het-s* heterokaryons producing only HET-s or HET-s* are tolerated since they contain a harmless HeLO domain. Additionally, homozygous *het-S/het-S* heterokaryons are viable since the HET-S is folded to hide the toxic HeLO domain. Heterozygotes *het-s/het-S* producing both HET-s* and HET-S are tolerated, as the rare transitions of HET-s* to HET-s/HET-S result in local cell death preventing prion propagation while the other cells that have remained HET-s*/HET-S remain viable [59]. In effect, this means that the HET-S protein suppresses the action of HET-s as a prion, due to this selection effect [59,60]. However, in homozygous *het-s/het-s* heterokaryons, HET-s* will eventually transform to the prion form and becomes HET-s/HET-s. Such a heterokaryon then triggers cell death if it encounters an individual that produces HET-S, whether that individual is *het-S/het-S* or *het-S/het-s*. Thus, such a system can create incompatibility between heterokaryons as well as between homokaryons, without triggering incompatibility within a heterokaryotic mycelium.

A second model from *P. anserina* may be found in the *het-V/het-V1* incompatibility system [61,62,63]. In this system, the *het-V* ‘allele’ is made of two genes: PaMt1, encoding a methyltransferase (writer), and PaMt2, encoding a protein with 2 domains MLKL (an executor, which causes cell death by membrane permeation) and TUDOR (reader: recognizes methylation status of target) [61]. The alternate “allele” *het-V1* is a null allele, with neither PaMt1 nor PaMt2. If the reader (TUDOR domain of PaMT2) senses a lack of methylation from the writer (methyltransferase of PaMT1) in certain target proteins, it directs cell death using the MLKL domain. Timely co-expression of reader and writer is analogous to restriction-modification in bacteria, with no cell death since all target molecules will receive the protective modification [10]. A confrontation between Podospora *het-V* and *het-V1* homokaryons results in an SI reaction, because the target proteins in *het-V1* cells that are not methylated, will subsequently be recognized by the reader (TUDOR) leading to cell death [61]. In this *het-V/het-V1* system, co-expression neutralizes the antagonistic alleles. Therefore, a heterokaryotic progeny *het-V/het-V1* is stable because the target proteins will be sufficiently methylated, preventing PaMt2 from triggering cell death. This *het-V/het-V1* heterokaryon would have a Het-V phenotype and be compatible with *het-V/het-V* heterokaryons, incompatibility would only result from such individual with Het-V phenotype to a *het-V1/het-V1* individual, highlighting the asymmetry in the *het-V/het-V1* system [62]. So, the *het-V/het-V1* system allows for SI to occur between heterokaryons (as well as between homokaryons), but heterokaryons are tolerated, an important condition of a basidiomycete model. We note that these properties result from the post-translational modification of proteins in the *het-V* system, and similar modifications to other cytoplasmic molecules would likely have similar results.

## 5. Outcomes of Traditional Models of Nonself Recognition

We now explore the implications for the different known nonself recognition systems for the typical basidiomycete lifecycle. We consider the lifecycle to include mating between unrelated homokaryons, and reciprocal nuclear migration between homokaryons that are sexually compatible as determined by unlinked mating loci. For the heterokaryotic stage, we assume heterokaryons are incompatible with other heterokaryons, but that nuclear migration from Buller matings is possible. The challenge is how we can explain that SI between two nuclei is avoided in a heterokaryotic cell, which would prevent their stable coexistence during the main stage of the basidiomycete lifecycle.

### 5.1. Sexual Compatibility Only When Somatic Compatibility Is Excluded (Allelic/Nonallelic Interactions)

Direct interactions between two protein products from two alleles, whether or not from homologous genes, are not compatible with the heterokaryotic lifecycle. The fusion of two distinct basidiomycete homokaryons would trigger nonself recognition directly inside the heterokaryon. The observation that homokaryons are generally sexually compatible rules out the possibility that mates are chosen based on compatibility at nonself recognition loci. This makes basidiomycete biology fundamentally different from the heterokaryotic *Neurospora tetrasperma* [54,55]. While both an allelic and non-allelic system would result in incompatibility between heterokaryons, the subsequent death within the heterokaryon rules out such a mechanism.

### 5.2. Differential Expression

Another possibility is the presence of an allelic or non-allelic system that is only expressed at the heterokaryon stage, using a mechanism opposite to the *tol* system of N. crassa. This seems like an attractive option since it allows the fusion of unrelated homokaryons, as they would not express the incompatible gene products, or not express part of the downstream pathway. This would allow for reciprocal nuclear migration. However, once nuclear migration is complete, the now dikaryotic cells would express the incompatible gene products, leading to the death of all dikaryotic cells. It could be possible that only one allele was expressed: allele-specific expression similar to genomic imprinting in plants/animals. Such a system would require precise regulation since a single locus that evades this imprinting would trigger cell death. Additionally, such a mechanism is not consistent with observations of nonself recognition in the fusion cells of sexually compatible homokaryons [43,44].

### 5.3. Post-Translational Modification

Perhaps a more promising model is found in a post-translational modification system. In such a system, heterokaryons are stable since the writers from the two nuclei modify all the target proteins in the cytoplasm, preventing the corresponding readers from triggering cell death. Such a system would be similar to the *het-V/het-V1* system of the *P. anserina* system discussed above in Section 4. The fusion of two heterokaryons would lead to nonself recognition, as long as the set of readers/writers differs between individuals. The fusion of heterokaryons will present a large number of unmodified targets, which will be detected by the corresponding reader molecule and trigger cell death. The fusion of two homokaryons would also present unmodified targets that would be recognized by the reader protein. This would also trigger nonself recognition and cell death. While at first, this seems incompatible with the basidiomycete lifecycle, detailed inspection has shown that cell death is found in sexually compatible homokaryons, although often slower than between heterokaryons [43,44]. It is possible that death of the fusion cell between homokaryons is common, but that occasional nuclear migration can escape prior to the death of the fusion cell. Since two homokaryons growing next to each other will have many hyphal fusions, only a small minority need to be successful for mating to occur due to the subsequent migration and mitosis.

### 5.4. Differential Expression of a Post-Translation Modification System

The above system could be modified by differential expression of the reader, writer, target, or a mix of these three. Limiting the expression of the target or the reader to the heterokaryon would prevent incompatibilities between homokaryons, but as discussed above, nonself recognition between homokaryons may still be compatible with the basidiomycete lifecycle. Furthermore, the targets of non-self-recognition are often essential genes, with other functional roles [27]. As such, the reduced expression would likely come with deleterious phenotypic side effects.

## 6. A Basidiomycete Scenario

Incorporating the previous information, we now consider a model of basidiomycete nonself recognition. We largely draw from the *het-V/het-V1* system, where allelic differences within a heterokaryon are tolerated but incompatibility occurs between heterokaryons based on post-translational modification of cytoplasmic proteins. We consider if such a reader/writer system is consistent with what we know of basidiomycete fungi. We include nuclear migration during mating as a potential explanation for two seemingly conflicting phenomena: Occurrence of cell death, and the formation of a heterokaryon. We first assume a first locus X which encodes the reader and the writer genes, with three alleles; X_A_, X_B,_ and X_C_. The genes of X_A_ both modify a set of proteins (Figure 3, light blue), as well as check their modification (Figure 3, light blue star). Likewise, the protein products of X_B_ and X_C_ perform similar functions but with different modifications or different targets.

Fusion between an AB (X_A_X_B_) and an AC (X_A_X_C_) heterokaryon will lead to SI. Inside the AB heterokaryon, for example, (Figure 3a), the X_A_X_B_ genes will modify targets. Fusion with the AC heterokaryon will then expose X_C_ to the unmodified targets inside the AB heterokaryon. Similar also for X_B_ and targets inside the AC heterokaryon. This nonself recognition would then be propagated to the cell death pathways and trigger cell death. A similar situation would occur with Buller matings (Figure 3b) and homokaryon-homokaryon matings (Figure 3c). However, in these cases, the unmodified targets of one individual would be recognized by the proteins of the other individual’s X locus. As the cell death process is not instantaneous, and the number of fusion cells is very high, the chance that at least one nucleus can begin migration before cell death is likely quite high.

This model of post-translation modification can explain the survival of heterozygous heterokaryons and the interaction between heterokaryons, but how do we deal with the mating reactions between homokaryons? One potential solution is that the nonself recognition phenotype is only expressed at the heterokaryotic stage, and so the fusion of dissimilar homokaryons would not lead to nonself recognition. When two homokaryons fuse, either the targets of the reader-writer, the reader-writer itself, or some downstream component would not be expressed. However, such differential expression between mono- and heterokaryons is insufficient to explain the success of Buller matings. Under a differential expression scenario, in such a Buller mating the heterokaryotic partner would recognize the homokaryon partner, and cell death would be triggered inside the fusion cell. Since Buller matings are a widely observed phenomenon in basidiomycetes, differential regulation does not seem to fit the observations.

An alternate explanation for a lack of observed incompatibility during homokaryon fusion and Buller matings, is nuclear migration leading to cytoplasmic dilution. When two homokaryons meet, assuming they have compatible mating loci, nuclei reciprocally migrate into each homokaryon. As only the nuclei migrate [47], they bring little to no cytoplasm with them, and both the reader-writer and its targets would remain in the original homokaryon. After migration, transcripts and thus proteins encoded by this new nucleus will begin to accumulate (Figure 4b). Assuming a resident nucleus A and a fertilizing nucleus B, the cytoplasm from A will initially contain the modified targets and the corresponding reader-writer from nucleus A. Transcription and translation of nucleus B will result in the accumulation of the reader-writer of nucleus B. If the accumulation of these proteins is gradual enough, the writer can timely and sufficiently modify the target and the signal produced by the respective readers from nucleus A and B will not reach above the threshold required to trigger cell death (Figure 4a,b, horizontal dotted line). Such a threshold, seen in programmed cell death in general [64,65], combined with nuclear migration, allows for the gradual co-expression of nucleus B inside the previously homokaryotic mycelia of A. Mature heterokaryons will accumulate sufficient target and reader/writer protein, such that fusion between heterokaryons could produce a signal that exceeds the threshold required for triggering cell death (Figure 4b).

Thus, the fact that two homokaryons are sexually compatible does not necessarily require the absence of nonself recognition. Instead, the rapid nuclear migration could allow for the escape of a small number of nuclei before cell death of the fusion cells. In fact, only a single nucleus needs to escape the nonself reaction for a successful mating reaction. Once a nucleus would make it past the boundaries of the fusion cell, subsequent migration does not bring cytoplasm along, meaning that further triggers of nonself recognition would be avoided during the migration process. Once the migrating nucleus reaches its terminus, *de novo* expression of the reader/writer would then result in gradual modification rather than cell death. We note here again that the death of fusion cells between sexually compatible homokaryons has been described [43,44,66]. The death of a fusion cell between homokaryon hyphae was described as being slower than the death between heterokaryotic hyphae but otherwise appeared similar [43]. In matings of *Agaricus bisporus*, for example, which does not have significant nuclear migration, pairings of homokaryons result in the formation of only a few localized successful heterokaryotic tufts (Figure 4c). This may indicate that although two homokaryons have compatible mating loci, most hyphal fusions do not result in heterokaryon formation. Interestingly, it has been noted that when auxotrophies were introduced into siblings of *Coprinopsis cinerea*, siblings with compatible mating alleles were always successful, but only a fraction of siblings could form heterokaryons if they had identical mating types, which prevents nuclear migration [67]. This lack of heterokaryon formation is a standard indication of SI in ascomycete genetics, and this finding in a model basidiomycete may indicate the similarity of the process if nuclear migration is inhibited. Although the death of fusion cells between homokaryons has been noted for over three decades, the implications of this have not been fully appreciated. What we term somatic incompatibility is a macromorphological phenomenon, and a lack of visible incompatibility does not necessarily mean that nonself recognition is absent, and *vice versa*, a macroscopic manifestation of incompatibility in a certain interaction does not necessarily imply that such an interaction cannot result in a stable heterokaryon.

## 7. Genetics of Proposed Model

As a rule, pairings of heterokaryotic wild isolates lead to SI [13,19,68,69]. Limited genetic studies have so far indicated that in basidiomycete species, nonself recognition is likely caused by 3–5 loci, some of which may be highly polymorphic in a population [16,17,70]. Although this number is lower than the 5–11 found in ascomycete species, the fact that a heterokaryon contains two genomes means that fewer polymorphic loci may be necessary to ensure a heterokaryon can distinguish itself from others. For example, consider heterokaryons produced from sibling matings from a single mushroom fruiting body. As these spores germinate and mate, they produce numerous heterokaryons from different sibling combinations. Although these siblings all originate from the same mushroom, and the different newly formed heterokaryons are thus closely related, if only a single locus was polymorphic between parents, then already 62.5% of heterokaryons would be incompatible (Table 1). If the parents were polymorphic at two loci, then 86% of offspring heterokaryons would be incompatible (1–0.365*0.365) if the two loci segregated independently. Further, the heterokaryotic state allows for interactions between multiple loci, since each heterokaryon contains two alleles, there can be up to 4 alleles present. The interaction patterns for polymorphisms at a triallelic locus result in 82% of randomly selected heterokaryons being incompatible due to only a single locus.

Clearly, this system allows for much more specificity than ascomycete systems, where the haploid genetics mean that each allele only confers incompatibility against 50% of other strains. This increased specificity may explain why fewer loci have been recovered in the basidiomycete systems studied to date [16,17,19]. If multiallelic loci are present, then polymorphisms at two triallelic loci would be sufficient to confer incompatibility between 97% of randomly selected heterokaryons (1–0.185*0.185) (Table 2). Assuming higher allelic states could even permit interactions between four alleles at a single locus, further reducing the compatible fraction. If the purpose of nonself recognition is indeed to prevent fusion between conspecific individuals, then very few loci may be necessary.

## 8. Conclusions

The particular lifecycle of basidiomycete fungi provides difficulties in understanding how nonself recognition can function, at least at a first glance. The long-term coexistence of two nuclear haplotypes excludes most of the so far known fungal nonself recognition mechanisms. Further, the successful mating between heterokaryons and homokaryons, Buller matings, shows that the expression of nonself recognition is not life-stage dependant. A major difference between ascomycete and basidiomycete fungi is the phenomenon of nuclear migration. Our model suggests that this migration is crucial for understanding how nonself recognition works, possibly through post-translation modification of a reader/writer system in a dose-dependent manner. Such a reader/writer system would allow for the coexistence of two nuclear haplotypes in a single cytoplasm, as well as matings between homokaryons and Buller matings. This limits the ability of parasitic cytoplasmic elements to spread at all life stages. Additionally, such reader/writer systems may act in a dominant fashion, and the use of a “common partner” nucleus (Worrall 1997) in genetic studies may not be as neutral as assumed. Further experiments are necessary to test if this model is compatible with the genetics and mechanism of SI in basidiomycete fungi.

## Figures and Tables

**Figure 1 microorganisms-09-01248-f001:**
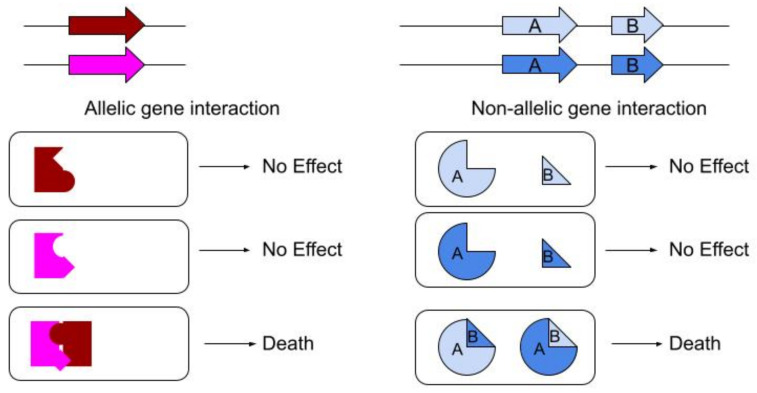
Traditional models for gene interactions leading to cell death from nonself recognition in ascomycete systems. The interactions can be based on direct gene product interactions, whether allelic (alleles of the same gene), or non-allelic (with alternate alleles of genetically linked genes). These systems act similarly upon fusion, but non-allelic systems differ during reproduction, where self-incompatible offspring result from the interaction produced from recombination between genes in a nonallelic locus. Note that non-allelic interactions are often asymmetric (e.g., only the combination of light blue A and dark blue B may be toxic, the interaction of dark blue A and light blue B may have no effect).

**Figure 2 microorganisms-09-01248-f002:**
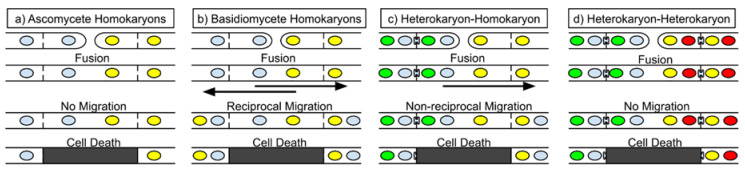
Nuclear migration permits mating in basidiomycetes despite the death of fusion cells. (**a**) The left panel shows the conventional ascomycete scenario, the fusion and subsequent cell death between two genetically different homokaryons; (**b**) shows the basidiomycete scenario when two different homokaryons meet and fuse, it generally leads to mating and reciprocal nuclear migration even though the fusion cell may die; (**c**) shows the outcome of a Buller mating, with unidirectional nuclear migration into the homokaryon individual, producing two different heterokaryons separated by a now-dead fusion cell; (**d**) When two distinct heterokaryons fuse, for example, mated basidiomycetes, it leads to nonself recognition and cell death like the others, but the lack of nuclear migration means that no further matings are produced. The scenario in (**d**) also applies for pseudohomothallic ascomycetes such as *Neurospora tetrasperma* or *Podospora anserina*.

**Figure 3 microorganisms-09-01248-f003:**
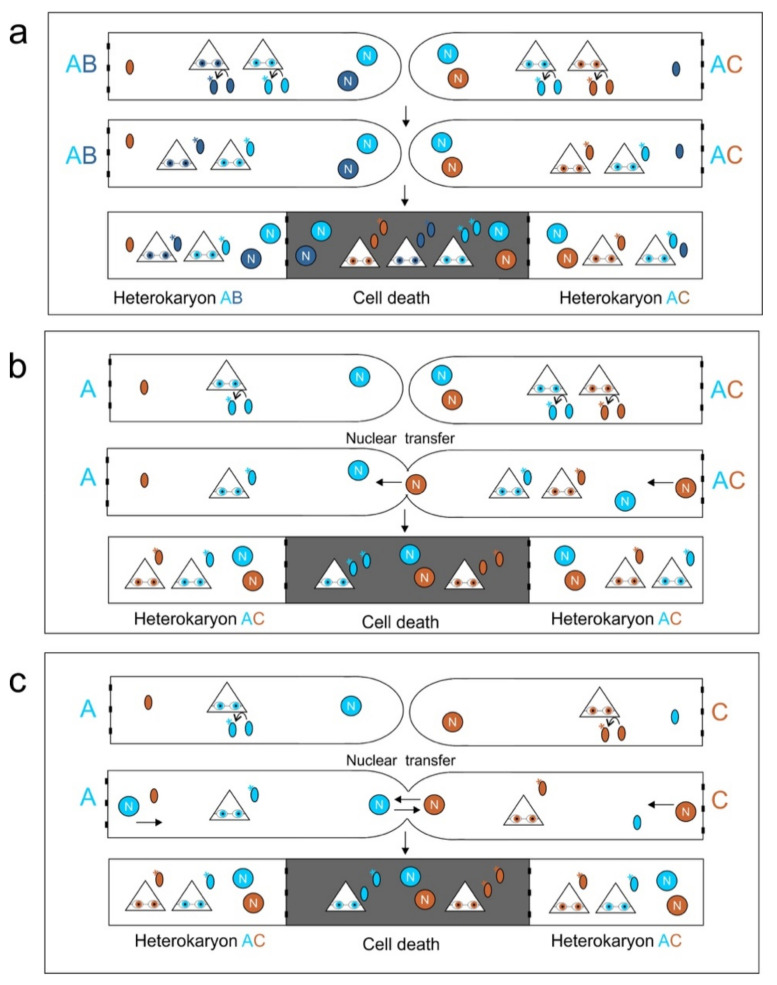
A basidiomycete scenario for a reader/writer involvement in nonself recognition. (**a**) interactions between heterokaryons. Each nucleus (circle with letter N) produces a reader/writer (triangle with glasses) that both modifies (asterisks) its target (oval), as well as monitors for the presence of an unmodified target. Each nucleus of each heterokaryon modifies its own target, but each leaves one target unmodified (e.g., brown targets in the AB heterokaryon). Following fusion of the heterokaryons, these unmodified targets are recognized by the alternate reader/writer leading to cell death. (**b**) Similar to described above, but now nuclear migration in the Buller mating from the AD heterokaryon into the homokaryon allows the escape of cell death by timely expression of the reader/writer in the resultant heterokaryon. (**c**) interactions between homokaryons lead to a mating reaction, as bidirectional nuclear migration allows the escape of the same cell death beyond the fused cell.

**Figure 4 microorganisms-09-01248-f004:**
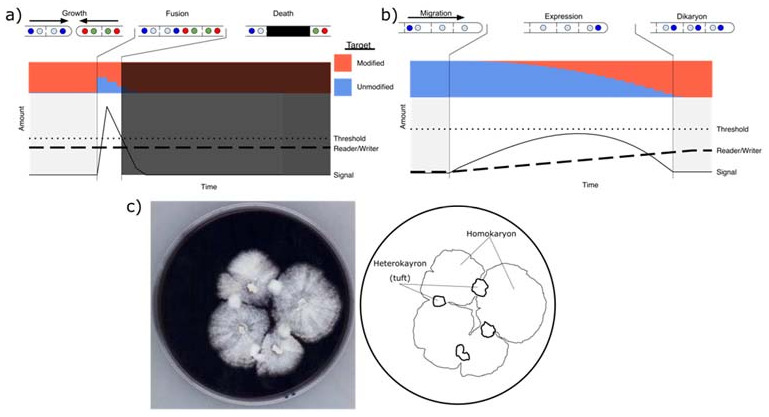
Dynamics of a reader/writer system during hyphal fusion, cytoplasmic mixing, and mating. (**a**) Graph of dynamics of any fusion involving cytoplasmic mixing, in this case between heterokaryotic basidiomycete mycelium. Dotted line shows the threshold above which cell death is triggered. Dark dashed line shows the concentration of the reader/writer of the newly resident blue nucleus. Solid line indicates the signal produced by the activity of the reader, which produces a signal proportional to the amount of unmodified target protein (Blue in the graph above). In this example, the reader has an activity five times higher than the writing activity, and the signal does not rise above the threshold. In this case, the high concentration of reader/writer protein immediately raises the signal above the threshold for cell death, based on the presence of unmodified target protein; (**b**) dynamics of the tip cell of a homokaryon during fertilization; (**c**) Mating reaction in *Agaricus bisporus*, a species with little nuclear migration, showing the relatively limited production of heterokaryons (“tufts”) in the interaction zone between homokaryons. (photo credit: Mushroom Group, WUR).

**Table 1 microorganisms-09-01248-t001:** Compatibility between 37.5% of randomly selected heterokaryons variable at a single biallelic locus. Squares with the letter C indicate compatible heterokaryons.

	A1/A1	A1/A2	A2/A1	A2/A2
A1/A1	C			
A1/A2		C	C	
A2/A1		C	C	
A2/A2				C

**Table 2 microorganisms-09-01248-t002:** Compatibility between 18.5% of randomly selected heterokaryons variable at a single triallelic locus. Squares with the letter C indicate compatible heterokaryons.

	A1/A1	A1/A2	A1/A3	A2/A1	A2/A2	A2/A3	A3/A1	A3/A2	A3/A3
A1/A1	C								
A1/A2		C		C					
A1/A3			C				C		
A2/A1		C		C					
A2/A2					C				
A2/A3						C		C	
A3/A1			C				C		
A3/A2						C		C	
A3/A3									C

## Data Availability

No data was generated.

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
