# Peer review of "Cytoplasmic Mixing, Not Nuclear Coexistence, Can Explain Somatic Incompatibility in Basidiomycetes"

_microorganisms, 2021, doi:10.3390/microorganisms9061248_

Round 1

Reviewer 1 Report

Review of “Cytoplasmic mixing, not nuclear coexistence, can explain somatic incompatibility in basidiomycetes” by Auxier et al., In this review, the authors extend molecular and genetic analyses on somatic incompatibility in ascomycete species to potential mechanisms in basidiomycete species.  Somatic incompatibility is widespread in both ascomycete and basidiomycete species.  However, as the authors point out, the life style of basidiomycete species as an extended heterokaryon formed as a result of monokaryotic fusion of mating compatible strains or by Buller’s mating, whereby a mating compatible nucleus from a heterokaryon migrates to a monokaryon upon fusion to establish a heterokaryon (di-mon mating). The review is thorough, in depth, thought provoking and was a pleasure to read. I have only minor comments.   

The hypotheses regarding potential mechanisms of somatic incompatibility in basidiomycetes comes from two observations. The first is the observation of killing during mating, either between monokaryons, or in di-mon matings, and is based on observations from the mid 1980’s. The second is the extension of work reported in a thesis on the het-V system in Podospora anserina, whereby death upon fusion between incompatible strains is mediated by an executioner protein and a protein that presumably methylates the executioner protein (writer protein). Fusion between strains with alternative het-V alleles results in death because, apparently, the executioner protein is not properly methylated. Two additions would strengthen the review. First, it would have been useful to have a micrograph of death responses during compatible matings in basidiomycetes to support these observations, or at least a more comprehensive discussion of what the data from these old papers actually shows.  Second, the thesis on het-V is not easily accessible to readers—so additional molecular/mechanistic information on the function of this system is needed for proper understanding of how this might work in basidiomycete species. It would be great to have access to the het-V publication (bioXriv?) for this review.

Minor suggestions:

Genus and species names should be italicized throughout.

References need minor editing.

Use basidiomycete “species” and ascomycete “species”, rather than ascomycetes and basidiomycetes. Perhaps not needed throughout, but parts of the review sound a bit colloquial due to this usage.

Reviewer 2 Report

The paper "Cytoplasmic mixing, not nuclear coexistence, can explain somatic incompatibility in basidiomycetes" discuss how self recognition may work in Basidiomycetes based on the known systems in Ascomycetes. It is a theoretical paper based on a summary of previous knowledge and presents no new data. I think the title is somewhat overstating the results as the paper is not really a full discussion of all possible systems of "nuclear coexistence" but only systems known from a different group of fungi before, or if you like is a direct consequence of the Basidiomycete system. The paper is generally well written but there is room for improvement, especially in the structure of the paper to make it more clear. There is for example a figure explaining the system in N. crassa, which is not as important for the paper, b
ut non for the systems in P. anserina. Which are more important to understand the rest of the paper.

There are two things I miss in the paper. The first one is not necessarily important but I think it would be good with an evolutionary aspect of these systems, as it is now it is purely genetic. What was the ancestral system in Dikarya? Is the system in P. anserina unique for this group in Ascomycetes? If so why this kind of system in Basidiomycetes and nothing completely new? The second that I think is more important is how to proceed from here? How can we get data to test this theory?

Specific comments:

L10. The first sentence of the abstract could be more straight forward and stronger. As it is now it seem to say that we know how it works in Basidiomycetes, but know it even better in Ascomycetes.

1. Introduction - Goes from fungi to more general back to fungi. I think it would be more clear is going simply from general to fungi, to Basidiomycetea.

L108. I do not find any explanation of the abbreviation TOL.

L174. tol or TOL?

L173-175. This sentence seem to start out as presenting a fairly general case but in the end it is the system of Nerospora that is presented.

L236-276. It would be nice with some illustrations explaining the systems in P. anserina. The text becomes quite difficult to follow.

L299. tol or TOL?

L413-415. But would not such a "leaky" system also allow for harmful elements to pass also?
